# Fractional-Order LCL Filters: Principle, Frequency Characteristics, and Their Analysis

**Junhua Xu** [1,*] [ID], **Ermeng Zeng** [1], **Xiaocong Li** [1,2,*], **Guopeng He** [1], **Weixun Liu** [1] and **Xuanren Meng** [3]

1   College of Electrical Engineering, Guangxi University, Nanning 530004, China; 18278509349@163.com (E.Z.); gpaangho@outlook.com (G.H.); 15739333823@163.com (W.L.)
2   College of Mechanical and Electrical and Quality Technology Engineering, Nanning University, Nanning 530200, China
3   Electric Power Research Institute of Guangxi Power Grid Corporation, Nanning 530023, China; a1789436351@hotmail.com
*   Correspondence: xjh@gxu.edu.cn (J.X.); lhtlht@gxu.edu.cn (X.L.)

**Abstract:** The fractional-order LCL filter, composed of two fractional-order inductors and one fractional-order capacitor, is a novel fractional-order π-type circuit introduced in recent years. Based on mathematical modeling, this article comprehensively studies the principles and frequency characteristics of fractional-order LCL filters. Five critical properties are derived and rigorously demonstrated. One of the most significant findings is that we identify the necessary and sufficient condition for resonance in fractional-order LCL filters when the sum of the orders of the fractional-order inductors and the fractional-order capacitor is equal to 2, which provides a theoretical foundation for effectively avoiding resonance in fractional-order LCL filters. The correctness of our theoretical derivation and analysis was confirmed through digital simulations. This study reveals that fractional-order LCL filters exhibit more versatile operational characteristics than traditional integer-order LCL filters, paving the way for broader application prospects.

**Keywords:** fractional-order capacitor; fractional-order inductor; fractional-order LCL filter; frequency characteristics; resonance

## 1. Introduction

The development of fractional calculus and the research on the implementation methods of fractional-order elements have opened up new approaches for the design, modeling, and analysis of circuits [1–11]. Fractional-order circuits have become an emerging interdisciplinary research area and have drawn much interest [5].

Fractional-order inductors and fractional-order capacitors [12,13] are the essential building elements of fractional-order circuits. The mathematical models of a fractional-order inductor and a fractional-order capacitor can be expressed as [6–8].

$$\begin{cases} L\frac{\mathrm{d}^{\alpha} i_{\mathrm{L}}}{\mathrm{d}t^{\alpha}} = u_{\mathrm{L}}, \\ C\frac{\mathrm{d}^{\beta} u_{\mathrm{C}}}{\mathrm{d}t^{\beta}} = i_{\mathrm{C}}, \end{cases} \tag{1}$$

where $L$ is the inductance of a fractional-order inductor, and $C$ is the capacitance of a fractional-order inductor. $\alpha$ and $\beta$ are the order of a fractional-order inductor and a fractional-order capacitor, respectively. $\alpha$ and $\beta$ satisfy $0 < \alpha, \beta < 2$. Particularly, when $\alpha = 1$ and $\beta = 1$, a fractional-order inductor and a fractional-order capacitor are traditional inductors and capacitors, respectively. $u_{\mathrm{L}}$ is the inductor voltage, $i_{\mathrm{L}}$ is the inductor current, $u_{\mathrm{C}}$ is the capacitor voltage, and $i_{\mathrm{C}}$ is the capacitor current.

From (1), we know that in addition to the traditional parameter (such as $L$ or $C$), the fractional-order element introduces an additional parameter, namely order, which makes the element and system more complex and more difficult to analyze, on the one hand, but,

on the other hand, it increases the flexibility and degree of freedom and brings in immense versatility towards design and application. For more information about the operating characteristics of fractional-order inductors and fractional-order capacitors, please refer to [3,6–8,12,13].

In recent years, many realizations and fabrication methods of fractional-order inductors and fractional-order capacitors have been reported [14–30], and the orders of some capacitors, such as super-capacitors, were verified as possibly being much further away from 1 by experiments [27]. At the same time, fractional-order inductors and fractional-order capacitors have gradually been applied to various traditional integer-order circuits, and counterpart fractional-order circuits are constructed, such as fractional-order chaotic circuits [3], fractional-order sinusoidal oscillators [4], fractional-order RLC/RC/RL circuits [6,7,10,11], fractional-order power electronic converters [8,9,31–36], etc.

The traditional integer-order LCL filter [37–39], which consists of two inductors, $L_1$ and $L_2$, and one capacitor, $C$, is a pivotal circuit extensively utilized in power electronic converters, such as in a grid-connected inverter, as shown in Figure 1a.

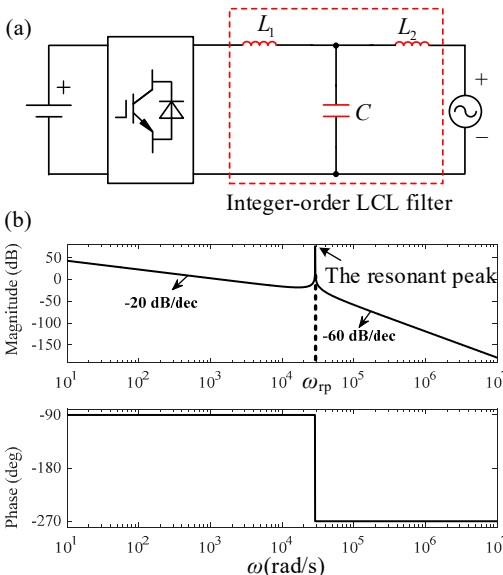

**Figure 1.** The main circuit topology and Bode plot of an integer-order LCL filter. (**a**) A grid-connected inverter with an integer-order LCL filter. (**b**) The Bode plot of an integer-order LCL filter.

Figure 1b displays the Bode plot of an integer-order LCL filter, where $\omega_{rp}$ denotes resonant frequency. When $\omega < \omega_{rp}$ and $\omega > \omega_{rp}$, the asymptotic slopes of the logarithmic magnitude-frequency characteristic curve of an integer-order LCL filter are $-20$ dB/dec and $-60$ dB/dec, respectively [39]. The integer-order LCL filter plays a key role in filtering high-order harmonics in the AC side voltage and improving the quality of the current injected into the grid [37–39]. Compared with L filters, integer-order LCL filters can achieve a better high-frequency harmonic attenuation effect. However, the integer-order LCL filter has an inherent resonant peak, which will destroy the stability of the system when the system is not designed and controlled properly [37–39].

A fractional-order capacitor is introduced to replace the integer-order capacitor in an integer-order LCL filter to form an LC$^\alpha$L filter [40]. Theoretical analysis and numerical simulations show that the LC$^\alpha$L filter provides a wider bandwidth to mitigate higher-order resonant frequencies than its integer-order counterpart. The circuit and mathematical model of a fractional-order LCL filter are preliminarily given in [41]. However, the detailed theoretical derivation and analysis of the frequency characteristics are missing.

This study employs Caputo's fractional calculus [1] to investigate the mathematical modeling and frequency characteristic analysis of fractional-order LCL filters. The aim is to expand the LCL filters from the integer-order domain to the fractional-order domain,

laying a theoretical groundwork for the engineering application of fractional-order LCL filters. The main contributions and originality of this research are summarized as follows:

(1) This paper pioneered a method for the theoretical analysis of the frequency characteristics of fractional-order LCL filters, summarized their five critical properties, and systematically revealed their principles and frequency characteristics.

(2) It is found that the necessary and sufficient condition for resonance in the magnitude-frequency characteristic curve of fractional-order LCL filters is that the sum of the orders of the fractional-order inductor and the fractional-order capacitor is equal to 2. This provides a theoretical basis for effectively avoiding resonance in fractional-order LCL filters.

(3) This paper fills the gap in the research on the frequency characteristics of general fractional systems with $2\alpha + \beta$-order (where $\alpha, \beta \in (0, 2)$).

The rest of this paper is organized as follows: the circuit and mathematical models of a fractional-order LCL filter are given in Section 2. In Section 3, the frequency characteristics of a fractional-order LCL filter are studied in depth, and five important properties of a fractional-order LCL filter are systematically presented. In Section 4, digital simulations are performed to verify the correctness of the theoretical derivation and analysis. Finally, conclusions are drawn in Section 5.

## 2. The Circuit and Mathematical Models of a Fractional-Order LCL Filter

By introducing the fractional-order inductors and the fractional-order capacitor to replace the inductors and the capacitor in an integer-order LCL filter, respectively, the main circuit of a fractional-order LCL filter can be obtained, as shown in Figure 2a, where the fractional-order LCL filter is used for the AC side filtering of a grid-connected inverter. $L_1$ denotes the inductance of the input side fractional-order inductor in units of H/ sec$^{(1-\alpha_1)}$, and its order is $\alpha_1$; $L_2$ denotes the inductance of the output side fractional-order inductor in units of H/ sec$^{(1-\alpha_2)}$, and its order is $\alpha_2$; $C$ denotes the capacitance of the fractional-order capacitor in units of F/ sec$^{(1-\beta)}$, and its order is $\beta$. All the value ranges of $\alpha_1$, $\alpha_2$, and $\beta$ are (0, 2). Obviously, a fractional-order LCL filter is the general form of an LCL filter, and an integer-order LCL filter is just the special case of a fractional-order LCL filter when $\alpha_1 = \alpha_2 = \beta = 1$. In addition, $u_{dc}$ is the DC voltage provided by the renewable energy generation or energy storage unit; $u_i$, $u_g$, and $u_C$ are the input voltage, the output voltage, and the fractional-order capacitor's terminal voltage of a fractional-order LCL filter, respectively. $i_1$, $i_2$, and $i_C$ are the input current, the output current, and the fractional-order capacitor's current of a fractional-order LCL filter, respectively. When a fractional-order LCL filter is used for the AC side filtering of a grid-connected inverter, $u_i$, $u_g$, and $i_2$ are also known as the AC side voltage of the inverter, the power grid voltage, and the current injected into the grid, respectively.

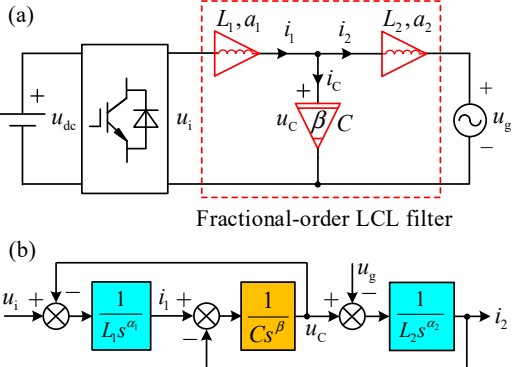

**Figure 2.** The main circuit topology and the model structure of a fractional-order LCL filter. (**a**) The main circuit topology of a fractional-order LCL filter. (**b**) The model structure of a fractional-order LCL filter.

According to the Kirchhoff voltage law and the Kirchhoff current law, the time domain mathematical model of a fractional-order LCL filter can be expressed as

$$
\begin{cases}
L_1 \dfrac{\mathrm{d}^{\alpha_1} i_1}{\mathrm{d}t^{\alpha_1}} = u_i - u_C, \\
C \dfrac{\mathrm{d}^{\beta} u_C}{\mathrm{d}t^{\beta}} = i_1 - i_2, \\
L_2 \dfrac{\mathrm{d}^{\alpha_2} i_2}{\mathrm{d}t^{\alpha_2}} = u_C - u_g.
\end{cases}
\tag{2}
$$

By performing the Laplace transform on (2), the *s*-domain expression of a fractional-order LCL filter can be obtained as

$$
\begin{cases}
L_1 s^{\alpha_1} i_1(s) = u_i(s) - u_C(s), \\
C s^{\beta} u_C(s) = i_1(s) - i_2(s), \\
L_2 s^{\alpha_2} i_2(s) = u_C(s) - u_g(s).
\end{cases}
\tag{3}
$$

According to (3), we obtain the model structure of a fractional-order LCL filter, as shown in Figure 2b.

According to (3) and Figure 2b, the output current expression of a fractional-order LCL filter can be obtained as

$$
i_2(s) = \frac{1}{L_1 L_2 C s^{\alpha_1 + \alpha_2 + \beta} + L_1 s^{\alpha_1} + L_2 s^{\alpha_2}} u_i(s) - \frac{L_1 C s^{\alpha_1 + \beta} + 1}{L_1 L_2 C s^{\alpha_1 + \alpha_2 + \beta} + L_1 s^{\alpha_1} + L_2 s^{\alpha_2}} u_g(s).
\tag{4}
$$

In order to simplify the analysis, let $\alpha_1 = \alpha_2 = \alpha$; then, (4) can be simplified as

$$
i_2(s) = \frac{1}{L_1 L_2 C s^{2\alpha + \beta} + (L_1 + L_2) s^{\alpha}} u_i(s) - \frac{L_1 C s^{\alpha + \beta} + 1}{L_1 L_2 C s^{2\alpha + \beta} + (L_1 + L_2) s^{\alpha}} u_g(s).
\tag{5}
$$

Generally, $u_g(s)$ is regarded as the disturbance of the system, so the transfer function from the input voltage $u_i(s)$ to the output current $i_2(s)$ can be obtained; that is, the transfer function of a fractional-order LCL filter is

$$
G_{gi}(s) = \frac{i_2(s)}{u_i(s)} = \frac{1}{L_1 L_2 C s^{\alpha}} \frac{1}{s^{\alpha + \beta} + A}.
\tag{6}
$$

where

$$
A = (L_1 + L_2)/(L_1 L_2 C), \ \alpha + \beta \in (0, 4).
\tag{7}
$$

From (6), we know that a fractional-order LCL filter is a fractional-order system with order $2\alpha + \beta$, where $\alpha, \beta \in (0, 2)$. Until now, no study has systematically introduced the frequency characteristics of such fractional-order systems.

According to (6), the frequency domain model of a fractional-order LCL filter can be expressed as

$$
G_{gi}(j\omega) = \frac{1}{L_1 L_2 C (j\omega)^{\alpha}} \frac{1}{(j\omega)^{\alpha + \beta} + A}.
\tag{8}
$$

Substituting the Euler equation

$$
(j\omega)^{\alpha} = \omega^{\alpha} e^{j\alpha\pi/2} = \omega^{\alpha} \cos\frac{\alpha\pi}{2} + j\omega^{\alpha} \sin\frac{\alpha\pi}{2}
$$

into (8), the frequency domain expression of a fractional-order LCL filter can be further expressed as

$$
G_{gi}(j\omega) = \frac{1}{L_1 L_2 C \omega^{\alpha} \left(\cos\frac{\alpha\pi}{2} + j\sin\frac{\alpha\pi}{2}\right)} \frac{1}{\omega^{\alpha + \beta} \cos\frac{(\alpha + \beta)\pi}{2} + A + j\omega^{\alpha + \beta} \sin\frac{(\alpha + \beta)\pi}{2}}.
\tag{9}
$$

Based on (9), the expressions of the magnitude-frequency characteristics and the phase-frequency characteristics of a fractional-order LCL filter can be obtained as

$$\left|G_{\text{gi}}(j\omega)\right| = \frac{1/(L_1 L_2 C \omega^\alpha)}{\sqrt{\left(\omega^{\alpha+\beta} + A \cos \frac{(\alpha+\beta)\pi}{2}\right)^2 + A^2 \sin^2 \frac{(\alpha+\beta)\pi}{2}}}. \tag{10}$$

$$\angle G_{\text{gi}}(j\omega) = -\arctan\left(\tan \frac{\pi\alpha}{2}\right) - \arctan \frac{\omega^{\alpha+\beta} \sin \frac{(\alpha+\beta)\pi}{2}}{\omega^{\alpha+\beta} \cos \frac{(\alpha+\beta)\pi}{2} + A}. \tag{11}$$

## 3. The Frequency Characteristics and Analysis of a Fractional-Order LCL Filter

### 3.1. The Resonance Characteristics of a Fractional-Order LCL Filter

In (10), when $\alpha + \beta \in (0,1] \cup [3,4)$, since $\cos[(\alpha+\beta)\pi/2] \geq 0$, with the angular frequency $\omega$ increasing from 0, the denominator of $\left|G_{\text{gi}}(j\omega)\right|$ increases monotonically; that is, $\left|G_{\text{gi}}(j\omega)\right|$ decreases monotonically. In this case, there is no resonance in the magnitude-frequency characteristics of a fractional-order LCL filter.

When $\alpha + \beta \in (1,3)$, $\cos[(\alpha+\beta)\pi/2] < 0$, let us define an intermediate variable

$$\omega_{\text{r}} = [-A \cos (\alpha+\beta)\pi/2]^{\frac{1}{\alpha+\beta}}, \quad \alpha + \beta \in (1,3), \tag{12}$$

namely, $\omega_{\text{r}}^{\alpha+\beta} = -A \cos[(\alpha+\beta)\pi/2]$.

Substituting the expression of $\omega_{\text{r}}$ into (10), we obtain

$$\left|G_{\text{gi}}(j\omega)\right| = \frac{1/(L_1 L_2 C \omega^\alpha)}{\sqrt{\left(\omega^{\alpha+\beta} - \omega_{\text{r}}^{\alpha+\beta}\right)^2 + A^2 \sin^2 \frac{(\alpha+\beta)\pi}{2}}}. \tag{13}$$
$$\alpha + \beta \in (1,3)$$

When the angular frequency satisfies

$$\omega = \omega_{\text{r}} = \left[-A \cos \frac{(\alpha+\beta)\pi}{2}\right]^{\frac{1}{\alpha+\beta}}, \quad \alpha + \beta \in (1,3), \tag{14}$$

$\left|G_{\text{gi}}(j\omega)\right|$ can be expressed as

$$\left|G_{\text{gi}}(j\omega_{\text{r}})\right| = \frac{1/(L_1 L_2 C \omega_{\text{r}}^\alpha)}{A|\sin[(\alpha+\beta)\pi/2]|}, \quad \alpha + \beta \in (1,3). \tag{15}$$

According to (15), we know that when $\alpha + \beta = 2$, due to $A \sin[(\alpha+\beta)\pi/2] = 0$, we have $\left|G_{\text{gi}}(j\omega_{\text{r}})\right| = \infty$. This means that the magnitude-frequency characteristics of a fractional-order LCL filter exhibit a resonance peak at $\omega = \omega_{\text{r}}$. Conversely, if the magnitude-frequency characteristics of a fractional-order LCL filter have a resonance peak when $\omega = \omega_{\text{r}}$, that is, $\left|G_{\text{gi}}(j\omega_{\text{r}})\right| = \infty$, it follows that $\alpha + \beta = 2$ must be satisfied. For ease of distinction, we denote the $\omega_{\text{r}}$ when $\alpha + \beta = 2$ as $\omega_{\text{rp}}$, which represents the resonant frequency.

From the above analysis, we discern that the fractional-order LCL filter possesses the following resonance property:

**Property 1.** *When $\alpha_1 = \alpha_2 = \alpha$, the necessary and sufficient condition for a fractional-order LCL filter to have a resonance peak is*
$$\alpha + \beta = 2. \tag{16}$$

Substituting $\alpha + \beta = 2$ into (12), the expression of the resonant frequency $\omega_{\text{rp}}$ can be obtained as follows:

$$\omega_{\text{rp}} = \sqrt{A} = \sqrt{(L_1 + L_2)/(L_1 L_2 C)}. \tag{17}$$

According to (17), we further obtain the following resonant frequency property of a fractional-order LCL filter:

**Property 2.** *When the magnitude-frequency characteristics of a fractional-order LCL filter have a resonant peak, that is, the condition of $\alpha + \beta = 2$ holds, the expression of the resonant frequency is $\omega_{rp} = \sqrt{(L_1 + L_2)/(L_1 L_2 C)}$. The resonant frequency $\omega_{rp}$ is only determined by the values of $L_1$, $L_2$, and $C$, and is independent of the order $\alpha$ of the fractional-order inductors and the order $\beta$ of the fractional-order capacitor.*

Property 1 reveals the condition for the existence of the resonance of a fractional-order LCL filter, essentially, and provides a straightforward criterion for determining whether a fractional-order LCL filter exhibits a resonant peak. For example, in traditional integer-order LCL filters, since both $\alpha$ and $\beta$ are 1, that is, $\alpha + \beta = 2$, their magnitude-frequency characteristics invariably exhibit a resonance peak, as illustrated in Figure 1b. Furthermore, Property 1 offers theoretical ground for effectively circumventing resonance in the design phase of a fractional-order LCL filter. When a fractional-order LCL filter manifests a resonance peak, it can readily lead to system instability [37–39]. For an integer-order LCL filter, additional passive or active dampers are commonly incorporated to eliminate resonance and enhance system stability against variations [37,38]. However, the former can incur additional losses, diminishing the attenuation capability of high-frequency harmonics, while the latter can complicate the design of the controller. In contrast, for a fractional-order LCL, we can effectively circumvent the resonance peak by judiciously selecting the order of fractional-order inductors and the fractional-order capacitor, such that $\alpha + \beta \neq 2$, thereby fundamentally enhancing system stability without adding dampers.

*3.2. The Corner Frequency and Logarithmic Magnitude-Frequency Characteristics of a Fractional-Order LCL Filter*

To analyze the corner frequency of a fractional-order LCL filter, according to the value range of $\alpha + \beta$, the following two intermediate variables of angular frequency are defined.

$$\omega_{t1} = \left| A \cos \frac{(\alpha + \beta)\pi}{2} \right|^{\frac{1}{\alpha + \beta}}, \ \alpha + \beta \in (0, 0.5] \cup [1.5, 2.5] \cup [3.5, 4]. \tag{18}$$

$$\omega_{t2} = \left| A \sin \frac{(\alpha + \beta)\pi}{2} \right|^{\frac{1}{\alpha + \beta}}, \ \alpha + \beta \in (0.5, 1.5) \cup (2.5, 3.5). \tag{19}$$

Figure 3 shows the waveforms of $\left| \cos(\alpha + \beta)\pi/2 \right|^{1/(\alpha + \beta)}$ in (18) and $\left| \sin(\alpha + \beta)\pi/2 \right|^{1/(\alpha + \beta)}$ in (19). From Figure 3, we know that when $\alpha + \beta \in (0, 0.5] \cup [1.5, 2.5] \cup [3.5, 4]$, there is $\omega_{t1} > \omega_{t2}$; when $\alpha + \beta \in (0.5, 1.5) \cup (2.5, 3.5)$, there is $\omega_{t1} < \omega_{t2}$.

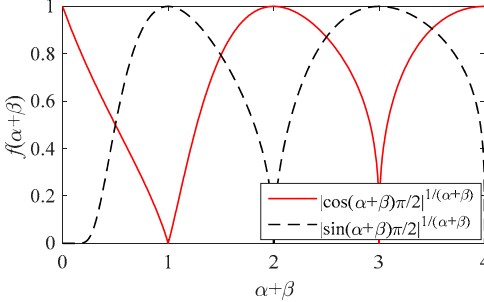

**Figure 3.** The waveforms of $\left| \cos(\alpha + \beta)\pi/2 \right|^{1/(\alpha + \beta)}$ and $\left| \sin(\alpha + \beta)\pi/2 \right|^{1/(\alpha + \beta)}$ versus $\alpha + \beta$.

Substituting (18) and (19) into (10), the magnitude-frequency characteristics of a fractional-order LCL filter can be expressed as

$$\left|G_{\mathrm{gi}}(j\omega)\right| = \frac{1/(L_1 L_2 C\omega^\alpha)}{\sqrt{\left(\omega^{\alpha+\beta} + \sigma\omega_{\mathrm{t1}}^{\alpha+\beta}\right)^2 + \omega_{\mathrm{t2}}^{2(\alpha+\beta)}}}, \tag{20}$$

where

$$\sigma = \begin{cases} 1, \alpha+\beta \in (0,1] \cup [3,4), \\ -1, \alpha+\beta \in (1,3). \end{cases} \tag{21}$$

(1) The analysis when $\alpha+\beta \in (0,0.5] \cup [1.5,2.5] \cup [3.5,4)$

According to the value range of $\omega$, (20) can be further expressed as

$$\left|G_{\mathrm{gi}}(j\omega)\right| = \begin{cases} \dfrac{1/\left(L_1 L_2 C\omega_{\mathrm{t1}}^{\alpha+\beta}\omega^\alpha\right)}{\sqrt{\left[(\omega/\omega_{\mathrm{t1}})^{\alpha+\beta}+\sigma\right]^2+(\omega_{\mathrm{t2}}/\omega_{\mathrm{t1}})^{2(\alpha+\beta)}}}, \omega << \omega_{\mathrm{t1}}, \\[3mm] \dfrac{1/\left[L_1 L_2 C\omega^{2\alpha+\beta}\right]}{\sqrt{\left[1+\sigma(\omega_{\mathrm{t1}}/\omega)^{\alpha+\beta}\right]^2+(\omega_{\mathrm{t2}}/\omega)^{2(\alpha+\beta)}}}, \omega >> \omega_{\mathrm{t1}}. \end{cases} \tag{22}$$

For the first formula of (22), when $\omega << \omega_{\mathrm{t1}}$, $(\omega/\omega_{\mathrm{t1}})^{\alpha+\beta} \approx 0$, considering that $\sigma^2 = 1$ always holds, so it can be simplified as

$$\left|G_{\mathrm{gi}}(j\omega)\right| \approx \frac{1/(L_1 L_2 C\omega^\alpha)}{\sqrt{\omega_{\mathrm{t1}}^{2(\alpha+\beta)} + \omega_{\mathrm{t2}}^{2(\alpha+\beta)}}} = \frac{1}{L_1 L_2 CA\omega^\alpha}, \omega << \omega_{\mathrm{t1}}. \tag{23}$$

According to (23), the logarithmic magnitude-frequency characteristic and its asymptotic slope of a fractional-order LCL filter when $\omega << \omega_{\mathrm{t1}}$ can be obtained as

$$L(\omega) \approx -20\lg(L_1 L_2 CA) - 20\alpha\lg\omega, \omega << \omega_{\mathrm{t1}}. \tag{24}$$

$$\frac{\mathrm{d}L(\omega)}{\mathrm{d}\lg\omega} \approx -20\alpha\mathrm{dB/dec}, \omega << \omega_{\mathrm{t1}}. \tag{25}$$

For the second formula of (22), when $\omega >> \omega_{\mathrm{t1}}$, $(\omega_{\mathrm{t1}}/\omega)^{\alpha+\beta} \approx 0$. Additionally, considering that there is $\omega_{\mathrm{t1}} > \omega_{\mathrm{t2}}$ when $\alpha+\beta \in (0,0.5] \cup [1.5,2.5] \cup [3.5,4)$, then $(\omega_{\mathrm{t2}}/\omega)^{\alpha+\beta} \approx 0$. So, it can be simplified as

$$\left|G_{\mathrm{gi}}(j\omega)\right| \approx \frac{1}{L_1 L_2 C\omega^{2\alpha+\beta}}, \omega >> \omega_{\mathrm{t1}}. \tag{26}$$

According to (26), the logarithmic magnitude-frequency characteristic and its asymptotic slope of a fractional-order LCL filter when $\omega >> \omega_{\mathrm{t1}}$ can be obtained as

$$L(\omega) = -20\lg(L_1 L_2 C) - 20(2\alpha+\beta)\lg\omega, \omega >> \omega_{\mathrm{t1}}. \tag{27}$$

$$\frac{\mathrm{d}L(\omega)}{\mathrm{d}\lg\omega} \approx -20(2\alpha+\beta)\mathrm{dB/dec}, \omega >> \omega_{\mathrm{t1}}. \tag{28}$$

(2) The analysis when $\alpha+\beta \in (0.5,1.5) \cup (2.5,3.5)$

When $\alpha+\beta \in (0.5,1.5) \cup (2.5,3.5)$, according to the value range of angular frequency, (20) can be further expressed as

$$\left|G_{\mathrm{gi}}(j\omega)\right| = \begin{cases} \dfrac{1/\left(L_1 L_2 C\omega^\alpha\omega_{\mathrm{t2}}^{\alpha+\beta}\right)}{\sqrt{\left[(\omega/\omega_{\mathrm{t2}})^{\alpha+\beta}+\sigma(\omega_{\mathrm{t1}}/\omega_{\mathrm{t2}})^{\alpha+\beta}\right]^2+1}}, \omega << \omega_{\mathrm{t2}}, \\[3mm] \dfrac{1/\left[L_1 L_2 C\omega^{2\alpha+\beta}\right]}{\sqrt{\left[1+\sigma(\omega_{\mathrm{t1}}/\omega)^{\alpha+\beta}\right]^2+(\omega_{\mathrm{t2}}/\omega)^{2(\alpha+\beta)}}}, \omega >> \omega_{\mathrm{t2}}. \end{cases} \tag{29}$$

For the first formula of (29), when $\omega << \omega_{t2}$, $(\omega/\omega_{t2})^{\alpha+\beta} \approx 0$, considering that $\sigma^2 = 1$ always holds, it can be simplified as

$$\left|G_{gi}(j\omega)\right| \approx \frac{1/(L_1L_2C\omega^\alpha)}{\sqrt{\omega_{t1}^{2(\alpha+\beta)} + \omega_{t2}^{2(\alpha+\beta)}}} = \frac{1}{L_1L_2CA\omega^\alpha}, \omega << \omega_{t2}. \tag{30}$$

According to (30), we obtain

$$L(\omega) \approx -20\lg(L_1L_2CA) - 20\alpha\lg\omega, \omega << \omega_{t2}, \tag{31}$$

$$\frac{dL(\omega)}{d\lg\omega} \approx -20\alpha\,dB/dec, \omega << \omega_{t2}, \tag{32}$$

For the second formula of (29), when $\omega >> \omega_{t2}$, $(\omega_{t2}/\omega)^{\alpha+\beta} \approx 0$. Additionally, considering that there is $\omega_{t2} > \omega_{t1}$ when $\alpha + \beta \in (0.5, 1.5) \cup (2.5, 3.5)$, then $(\omega_{t1}/\omega)^{\alpha+\beta} \approx 0$. So, it can be simplified as

$$\left|G_{gi}(j\omega)\right| \approx \frac{1}{L_1L_2C\omega^{2\alpha+\beta}}, \omega >> \omega_{t2}. \tag{33}$$

According to (33), we obtain

$$L(\omega) \approx -20\lg(L_1L_2C) - 20(2\alpha + \beta)\lg\omega, \omega >> \omega_{t2}, \tag{34}$$

$$\frac{dL(\omega)}{d\lg\omega} \approx -20(2\alpha + \beta)dB/dec, \omega >> \omega_{t2}. \tag{35}$$

(3) A summary of the corner frequency and logarithmic magnitude-frequency characteristic

Drawing from the aforementioned theoretical derivation, we can deduce that when $\alpha + \beta \in (0, 0.5] \cup [1.5, 2.5] \cup [3.5, 4)$, the corner frequency of the fractional-order LCL filter is $\omega_{t1}$, and when $\alpha + \beta \in (0.5, 1.5) \cup (2.5, 3.5)$, the corner frequency is $\omega_{t2}$. $\omega_{t1}$ and $\omega_{t2}$ can be condensed into the following expression for the corner frequency.

$$\omega_t = \begin{cases} \omega_{t1} = \left|A\cos\frac{(\alpha+\beta)\pi}{2}\right|^{\frac{1}{\alpha+\beta}}, \alpha + \beta \in (0, 0.5] \cup [1.5, 2.5] \cup [3.5, 4), \\ \omega_{t2} = \left|A\sin\frac{(\alpha+\beta)\pi}{2}\right|^{\frac{1}{\alpha+\beta}}, \alpha + \beta \in (0.5, 1.5) \cup (2.5, 3.5). \end{cases} \tag{36}$$

Furthermore, to facilitate discussions, we define the frequency ranges as the low-frequency band for $\omega << \omega_t$ and the high-frequency band for $\omega >> \omega_t$. By comparing (30)–(35) with (23)–(28), we observe that the expressions for the magnitude-frequency characteristics of the fractional-order LCL filter are identical when $\alpha + \beta \in (0, 0.5] \cup [1.5, 2.5] \cup [3.5, 4)$ and $\alpha + \beta \in (0.5, 1.5) \cup (2.5, 3.5)$. The primary distinction lies in their corner frequency expressions. In summary, the logarithmic magnitude-frequency characteristics of fractional-order LCL filters can be summarized as follows.

**Property 3.** *When $\omega << \omega_t$, the asymptotic slope of the logarithmic magnitude-frequency characteristic in the low-frequency band of a fractional-order LCL filter is $-20\,\alpha dB/dec$; when $\omega >> \omega_t$, the asymptotic slope of the logarithmic magnitude-frequency characteristic in the high-frequency band is $-20(2\alpha + \beta)dB/dec$.*

Property 3 is crucial for analyzing the filtering performance of fractional-order LCL filters. From Property 3, it can be observed that the asymptotic slope of the logarithmic magnitude-frequency characteristics of a fractional-order LCL filter in the low-frequency band is solely determined by the order $\alpha$ of the fractional-order inductors and is independent of the order $\beta$ of the fractional-order capacitor; in the high-frequency band, the

asymptotic slope is affected by both $\alpha$ and $\beta$. Considering that $\alpha, \beta \in (0, 2)$, we know that the asymptotic slope range of the logarithmic magnitude-frequency characteristics of a fractional-order LCL filter in the low-frequency band and high-frequency band is $(0\,\text{dB/dec}, -40\,\text{dB/dec})$ and $(0\,\text{dB/dec}, -120\,\text{dB/dec})$, respectively.

According to (36), we find that the corner frequency of a fractional-order LCL filter is not only affected by the values of $L_1$, $L_2$ and $C$ but also related to the order $\alpha$ of the fractional-order inductors and the order $\beta$ of the fractional-order capacitor. Additionally, when $\alpha + \beta = 2$, there is $\omega_{\text{rp}} = \omega_{\text{t1}}$, which means that the resonant frequency is equal to the corner frequency. In particular, when $\alpha = \beta = 1$, according to (25), (28), and (36), we obtain the corner frequency of an integer-order LCL filter as $\omega_\text{t} = \sqrt{A} = \sqrt{(L_1 + L_2)/(L_1 L_2 C)}$, and the asymptotic slopes of the low-frequency and high-frequency bands are $-20\,\text{dB/dec}$ and $-60\,\text{dB/dec}$, respectively.

*3.3. The Phase-Frequency Characteristics of a Fractional-Order LCL Filter*

It is known to all that for a complex number $z = x + jy$, $(z \neq 0)$, the value range of $-\pi < \arg z \leq \pi$ is generally called the principal argument of $z$, and there is

$$\arg z = \begin{cases} \arctan(y/x), & z \text{ is in the first and fourth quadrants,} \\ \pi + \arctan(y/x), & z \text{ is in the second quadrant,} \\ -\pi + \arctan(y/x), & z \text{ is in the third quadrant,} \end{cases} \tag{37}$$

where $-\pi/2 < \arctan(y/x) < \pi/2$.

From (37), the first term of (11) can be further expressed as

$$\arctan[\tan(\pi\alpha/2)] = \begin{cases} \pi\alpha/2, & \alpha \in (0, 1], \\ (\pi\alpha/2 - \pi) + \pi = \pi\alpha/2, & \alpha \in (1, 2), \end{cases} \tag{38}$$

which means $\arctan[\tan(\pi\alpha/2)] = \pi\alpha/2$ always holds when $\alpha \in (0, 2)$.

To facilitate the derivation, let the center frequency

$$\omega_\text{o} = A^{1/(\alpha+\beta)} = \sqrt[\alpha+\beta]{(L_1 + L_2)/(L_1 L_2 C)}, \tag{39}$$

then, the (11) can be further simplified as

$$\angle G_{\text{gi}}(j\omega) = -\frac{\pi\alpha}{2} - \arctan\frac{\sin\frac{(\alpha+\beta)\pi}{2}}{\cos\frac{(\alpha+\beta)\pi}{2} + (\omega_\text{o}/\omega)^{\alpha+\beta}}. \tag{40}$$

When $\omega \ll \omega_\text{o}$, considering the characteristics of the sin and cos functions, the following relationship always holds.

$$\sin\frac{(\alpha+\beta)\pi}{2} \ll \cos\frac{(\alpha+\beta)\pi}{2} + (\omega_\text{o}/\omega)^{\alpha+\beta},$$

then, there is

$$-\arctan\frac{\sin\frac{(\alpha+\beta)\pi}{2}}{\cos\frac{(\alpha+\beta)\pi}{2} + (\omega_\text{o}/\omega)^{\alpha+\beta}} \approx 0, \ \omega \ll \omega_\text{o}.$$

Therefore, the phase-frequency characteristics of a fractional-order LCL filter when $\omega \ll \omega_\text{o}$ are

$$\angle G_{\text{gi}}(j\omega) \approx -\pi\alpha/2, \ \omega \ll \omega_\text{o}. \tag{41}$$

When $\omega \gg \omega_\text{o}$, $\omega_\text{o}/\omega \approx 0$, so (40) can be further simplified as

$$\angle G_{\text{gi}}(j\omega) \approx -\frac{\pi\alpha}{2} - \arctan\left[\tan\frac{(\alpha+\beta)\pi}{2}\right], \ \omega \gg \omega_\text{o}. \tag{42}$$

According to (37), the second term in (42) can be further expressed as

$$\arctan\left[\tan\frac{(\alpha+\beta)\pi}{2}\right] = \begin{cases} \frac{(\alpha+\beta)\pi}{2}, & \alpha+\beta \in (0,2], \\ \frac{(\alpha+\beta)\pi}{2} - 2\pi, & \alpha+\beta \in (2,4). \end{cases} \tag{43}$$

By synthesizing (42) and (43), the phase-frequency characteristics of a fractional-order LCL filter when $\omega >> \omega_o$ can be obtained as

$$\angle G_{gi}(j\omega) \approx \begin{cases} -\pi(\alpha+\beta/2), & \alpha+\beta \in (0,2], \omega >> \omega_o, \\ -\pi(\alpha+\beta/2) + 2\pi, & \alpha+\beta \in (2,4), \omega >> \omega_o. \end{cases} \tag{44}$$

Based on (42) and (43), the phase-frequency characteristics of a fractional-order LCL filter can be summarized as follows.

**Property 4.** *When $\omega << \omega_o$, the phase of a fractional-order LCL filter is $-\pi\alpha/2$; when $\omega >> \omega_o$, if $\alpha + \beta \in (0,2]$, the phase is $-(\alpha + \beta/2)\pi$. If $\alpha + \beta \in (2,4)$, the phase is $-(\alpha + \beta/2)\pi + 2\pi$.*

According to Property 4, it can be observed that when $\omega << \omega_o$, the phase of the fractional-order LCL filter is solely determined by the order $\alpha$ of the fractional-order inductor, and it is independent of the order $\beta$ of the fractional-order capacitor. Conversely, when $\omega >> \omega_o$, the phase is influenced by both $\alpha$ and $\beta$.

*3.4. The Phase Crossover Frequency and Gain Margin of a Fractional-Order LCL Filter*

It is well known that the frequency at which the open-loop phase-frequency characteristic intersects the horizontal line $-\pi$ is commonly referred to as the phase crossover frequency, denoted as $\omega_g$. According to (40), we know that the intersection frequency $\omega_g$ should satisfy the following equation for a fractional-order LCL filter.

$$\angle G_{gi}(j\omega_g) = -\frac{\pi\alpha}{2} - \arctan\frac{\sin\frac{(\alpha+\beta)\pi}{2}}{\cos\frac{(\alpha+\beta)\pi}{2} + (\omega_o/\omega_g)^{\alpha+\beta}} = -\pi. \tag{45}$$

By solving (45), we obtain

$$\omega_g^{\alpha+\beta} = \frac{A\sin[(1-\alpha/2)\pi]}{-\sin[(\alpha+\beta/2)\pi]}. \tag{46}$$

Given that $0 < \alpha < 2$, we have $0 < (1-\alpha/2)\pi < \pi$, so the numerator of (46) satisfies $A\sin[(1-\alpha/2)\pi] > 0$. Considering the domains of $\alpha$ and $\beta$, the denominator of (46) satisfies $-\sin[(\alpha+\beta/2)\pi] > 0$, if and only if $\alpha + \beta/2 \in (1,2)$. This means the right side of (46) is greater than 0, indicating that (46) has a real number solution, implying the presence of a phase crossover frequency in the phase-frequency characteristic of a fractional-order LCL filter. To sum up, the phase crossover frequency characteristics of a fractional-order LCL filter can be summarized as follows.

**Property 5.** *If and only if $\alpha + \beta/2 \in (1,2)$, the phase-frequency characteristics of a fractional-order LCL filter has a phase crossover frequency $\omega_g$, and the expression of $\omega_g$ is*

$$\omega_g = \sqrt[\alpha+\beta]{\frac{A\sin[(1-\alpha/2)\pi]}{-\sin[(\alpha+\beta/2)\pi]}}. \tag{47}$$

In particular, when $\alpha = \beta = 1$, there is $\omega_g = \sqrt{A} = \sqrt{(L_1 + L_2)/(L_1 L_2 C)} = \omega_{rp}$; that is, the phase crossover frequency is equal to the resonant frequency.

When $\alpha + \beta/2 \in (1,2)$, the gain margin of a fractional-order LCL filter and its logarithmic expression can be obtained as

$$K_g = \frac{1}{|G_{gi}(j\omega_g)|} = L_1 L_2 C \omega_g^\alpha \sqrt{\omega_g^{2(\alpha+\beta)} + 2A\omega_g^{\alpha+\beta} \cos \frac{(\alpha+\beta)\pi}{2} + A^2}. \tag{48}$$

$$20K_g = -20\lg|G_{gi}(j\omega_g)| = -20\lg \frac{1/L_1 L_2 C \omega_g^\alpha}{\sqrt{\omega_g^{2(\alpha+\beta)} + 2A\omega_g^{\alpha+\beta} \cos \frac{(\alpha+\beta)\pi}{2} + A^2}}. \tag{49}$$

*3.5. The Gain Crossover Frequency and Phase Margin of a Fractional-Order LCL Filter*

As is widely known, the frequency $\omega_c$ satisfies $|G_{gi}(j\omega_c)| = 1$ or $20\lg|G_{gi}(j\omega_c)| = 0$ is usually referred to as the phase crossover frequency. According to (10), we know that the gain crossover frequency $\omega_c$ of a fractional-order LCL filter should satisfy the following equation.

$$\frac{1/(L_1 L_2 C \omega_c^\alpha)}{\sqrt{\left(\omega_c^{\alpha+\beta} + A \cos \frac{(\alpha+\beta)\pi}{2}\right)^2 + A^2 \sin^2 \frac{(\alpha+\beta)\pi}{2}}} = 1. \tag{50}$$

By sorting out Equation (50), we obtain

$$\omega_c^{2(2\alpha+\beta)} + 2A\omega_c^{3\alpha+\beta} \cos \frac{(\alpha+\beta)\pi}{2} + A^2 \omega_c^{2\alpha} - \frac{A^2}{(L_1+L_2)^2} = 0. \tag{51}$$

The above formula is a highly complex nonlinear equation, making it challenging to derive an analytical solution for the gain crossover frequency $\omega_c$ of a fractional-order LCL filter. In order to obtain the trend graph of the change in $\omega_c$ with the order of the fractional-order inductors and the order of the fractional-order capacitor, we successively used the fsolve function, the particle swarm optimization algorithm [42], and the differential evolution algorithm [43] to solve (49) based on MATLAB 2018a software, and we found that the differential evolution algorithm has the best solution effect and relatively stable results.

Figure 4 shows a change diagram of the digital solution of $\omega_c$ obtained by the differential evolution algorithm versus $\alpha - \beta$ plane when $\alpha, \beta \in (0.8, 1.2)$. The main circuit parameters of a fractional-order LCL filter are as follows: $L_1 = 600 \,\mu\text{H}/\sec^{(1-\alpha)}$, $L_2 = 150 \,\mu\text{H}/\sec^{(1-\alpha)}$, $C = 10 \,\mu\text{F}/\sec^{(1-\beta)}$.

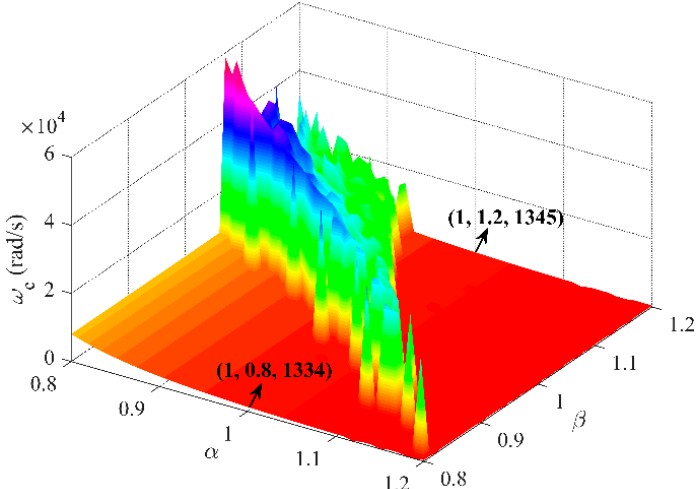

**Figure 4.** The change diagram of the digital solution of $\omega_c$ obtained by the differential evolution algorithm versus $\alpha - \beta$ plane.

From Figure 4, we observe that as the combination of $\alpha$ and $\beta$ approaches $\alpha + \beta = 2$, there is a notable increase in the value of $\omega_c$. This is attributed to the presence of a resonance peak in the fractional-order LCL filter when $\alpha + \beta = 2$, which makes the logarithmic magnitude-frequency characteristic curve cross the 0 dB line many times. This phenomenon will be further verified in the following digital simulations.

Additionally, according to Property 3 and Figure 4, we know the following:

(1) Except for the case of $\alpha + \beta = 2$, as the order $\alpha$ of the fractional-order inductors increases, the asymptotic slope of the logarithmic magnitude-frequency characteristic curve in the low-frequency band increases accordingly, and the gain crossover frequency decreases.

(2) Considering that the order $\beta$ of the fractional-order capacitor does not affect the low-frequency asymptotic slope of the logarithmic magnitude-frequency characteristic curve of a fractional-order LCL filter, altering the order $\beta$ does not alter the gain crossover frequency, except in the case of $\alpha + \beta = 2$.

The formula for calculating the phase margin of a fractional-order LCL filter is the same as that of an integer-order LCL filter, namely

$$\gamma = 180° + \angle G_{gi}(j\omega_c). \tag{52}$$

## 4. Simulation Results and Analysis

To validate the theoretical derivation, analyze the frequency characteristics of fractional-order LCL filters, and investigate the impact of the order of the fractional-order inductors and the fractional-order capacitor on these characteristics, a digital model of a fractional-order LCL filter is established and simulated using MATLAB software. The main circuit parameters of the fractional-order LCL filter are consistent with those in Figure 4, and the order of the fractional-order inductors and the fractional-order capacitor will be specified later.

### 4.1. The Frequency Characteristic Simulation Curves of a Fractional-Order LCL Filter

(1) The frequency characteristic curves when $\alpha$ is constant and $\beta$ changes

Simulation condition I: Set the order $\alpha$ of the fractional-order inductors to 0.8, 1.0, and 1.2, respectively. For each case, increase the order $\beta$ of the fractional-order capacitor from 0.6 to 1.4 in 0.2 intervals.

The Bode plots of a fractional-order LCL filter based on Simulation condition I, as shown in Figure 5, and the corresponding frequency characteristic indicators are summarized in Table 1. In this table, NaN indicates the absence of such indicators, and Inf denotes infinite indicators.

(2) The frequency characteristic curves when $\beta$ is constant and $\alpha$ changes

Simulation condition II: Set the order $\beta$ of the fractional-order capacitor to 0.8, 1.0, and 1.2, respectively. For each case, increase the order $\alpha$ of the fractional-order inductor from 0.6 to 1.4 in 0.2 intervals.

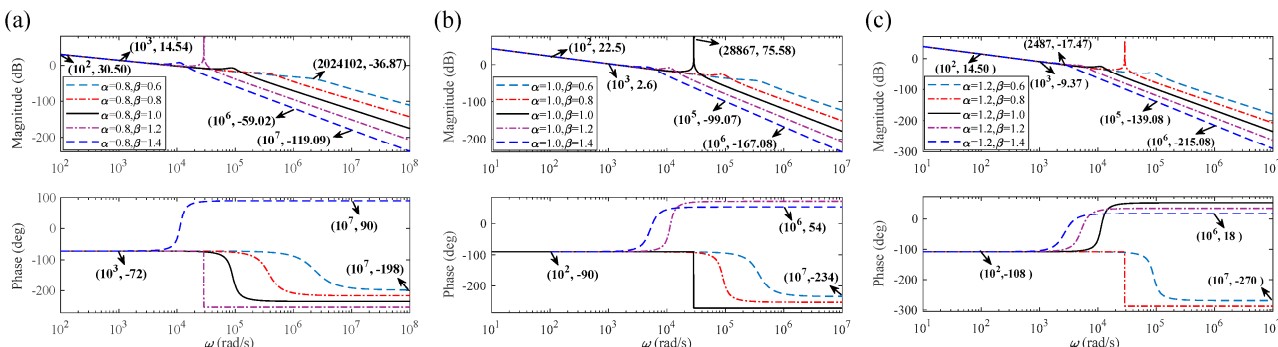

**Figure 5.** The frequency characteristic curves of a fractional-order LCL Filter when $\alpha$ is constant and $\beta$ changes. (**a**) The frequency characteristics when $\alpha = 0.8$. (**b**) The frequency characteristics when $\alpha = 1.0$. (**c**) The frequency characteristics when $\alpha = 1.2$.

**Table 1.** The frequency characteristic indicators of a fractional-order LCL when $\alpha$ is constant and $\beta$ changes.

| Serial Number | $\alpha$ | $\beta$ | $\omega_g$ (rad/s) | $20\lg K_g$ (dB) | $\omega_c$ (rad/s) | $\gamma$ (°) | $\omega_t$ (rad/s) |
|---|---|---|---|---|---|---|---|
| 1 | 0.8 | 0.6 | 5,257,083 | 53.39 | 8059 | 107.98 | 2,024,102 |
| 2 | 0.8 | 0.8 | 508,310 | 28.80 | 8075 | 107.93 | 329,599 |
| 3 | 0.8 | 1.0 | 98,864 | 9.06 | 8186 | 107.76 | 87,883 |
| 4 | 0.8 | 1.2 | 28,867 | −79.03 | 33,194 | −72.00 | 28,867 |
| 5 | 0.8 | 1.4 | NaN | Inf | 13,749 | 241.70 | 11,092 |
| 6 | 1.0 | 0.6 | 429,574 | 47.38 | 1334 | 90.00 | 329,598 |
| 7 | 1.0 | 0.8 | 92,922 | 27.10 | 1334 | 89.99 | 87,883 |
| 8 | 1.0 | 1.0 | 28,867 | −75.58 | 29,510 | 90.00 | 28,867 |
| 9 | 1.0 | 1.2 | NaN | Inf | 1345 | 90.16 | 11,092 |
| 10 | 1.0 | 1.4 | NaN | Inf | 1379 | 91.43 | 4772 |
| 11 | 1.2 | 0.6 | 87,883 | 45.96 | 402 | 71.99 | 87,882 |
| 12 | 1.2 | 0.8 | 28,867 | −57.73 | 28,950 | −108.0 | 28,867 |
| 13 | 1.2 | 1.0 | NaN | Inf | 402 | 72.01 | 11,092 |
| 14 | 1.2 | 1.2 | NaN | Inf | 402 | 72.07 | 4772 |
| 15 | 1.2 | 1.4 | NaN | Inf | 403 | 72.33 | 2487 |

The Bode plots of a fractional-order LCL filter, as shown in Figure 6, and the corresponding frequency characteristic indicators are summarized in Table 2.

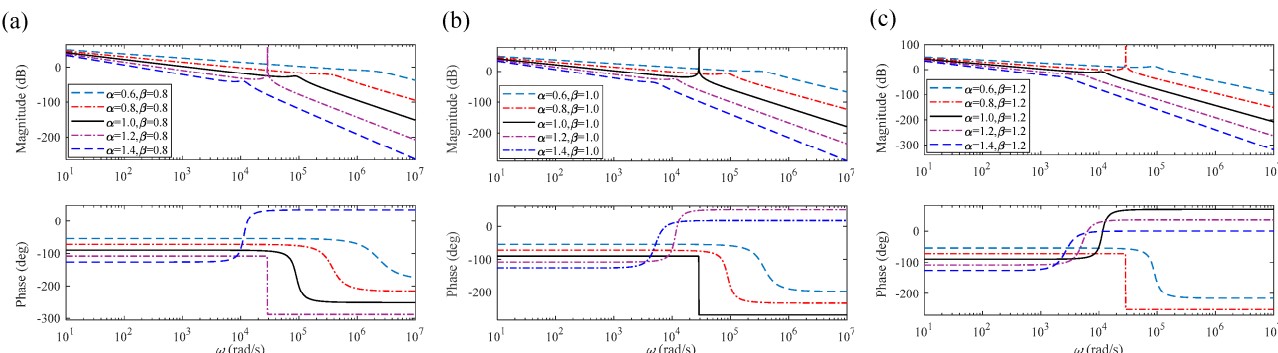

**Figure 6.** The frequency characteristic curves of a fractional-order LCL Filter when $\beta$ is constant and $\alpha$ changes. (**a**) The frequency characteristics when $\beta = 0.8$. (**b**) The frequency characteristics when $\beta = 1.0$. (**c**) The frequency characteristics when $\beta = 1.2$.

**Table 2.** The frequency characteristic indicators of a fractional-order LCL when $\beta$ is constant and $\alpha$ changes.

| Serial Number | $\alpha$ | $\beta$ | $\omega_g$ (rad/s) | $20\lg K_g$ (dB) | $\omega_c$ (rad/s) | $\gamma$ (°) | $\omega_t$ (rad/s) |
|---|---|---|---|---|---|---|---|
| 1 | 0.6 | 0.8 | 19,437,846 | 50.51 | 16,538 | 124.85 | 2,024,630 |
| 2 | 0.8 | 0.8 | 508,310 | 28.80 | 8075 | 107.93 | 329,599 |
| 3 | 1.0 | 0.8 | 92,922 | 27.10 | 1334 | 89.99 | 87,883 |
| 4 | 1.2 | 0.8 | 28,867 | −57.73 | 28,950 | −108.0 | 28,867 |
| 5 | 1.4 | 0.8 | NaN | Inf | 171 | 54.00 | 11,092 |
| 6 | 0.6 | 1.0 | 686,670 | 13.13 | 368,962 | 56.77 | 329,598 |
| 7 | 0.8 | 1.0 | 98,864 | 9.06 | 8186 | 107.76 | 87,882 |
| 8 | 1.0 | 1.0 | 28,867 | −75.58 | 29,510 | −90.00 | 28,867 |
| 9 | 1.2 | 1.0 | NaN | Inf | 402 | 72.01 | 11,092 |
| 10 | 1.4 | 1.0 | NaN | Inf | 171 | 54.01 | 4772 |
| 11 | 0.6 | 1.2 | 107,916 | −7.69 | 133,810 | −19.98 | 87,883 |
| 12 | 0.8 | 1.2 | 28,867 | −93.42 | 33,192 | −72.00 | 28,867 |
| 13 | 1.0 | 1.2 | NaN | Inf | 1345 | 90.16 | 11,092 |
| 14 | 1.2 | 1.2 | NaN | Inf | 402 | 72.07 | 4771 |
| 15 | 1.4 | 1.2 | NaN | Inf | 170.7 | 54.04 | 2487 |

*4.2. The Analysis of the Frequency Characteristics of a Fractional-Order LCL Filter*

From Figures 5 and 6, as well as Tables 1 and 2, we can observe that:

(1) When $\alpha + \beta = 2$, for instance, the combinations of $\alpha$ and $\beta$ are (0.8, 1.2), (1.0, 1.0), and (1.2, 0.8), the magnitude-frequency characteristic curves of these fractional-order LCL filters cross the 0 dB line multiple times and exhibit a resonance peak at the resonant frequency. Moreover, the phase-frequency characteristic curves make a jump of $-180°$ at the resonant frequency. For example, when the combination of $\alpha$ and $\beta$ is (0.8, 1.2), the phase-frequency characteristic curve makes a jump from $-72°$ to $-252°$. Furthermore, it can be observed that the resonant frequency $\omega_{\mathrm{rp}}$ is solely determined by $L_1$, $L_2$ and $C$ and is independent of $\alpha$ and $\beta$. In this scenario, the theoretically calculated resonant frequency is

$$\omega_{\mathrm{rp}} = \sqrt{(L_1 + L_2)/(L_1 L_2 C)} \approx 28{,}867.5 \text{ rad/s,}$$

and the measured resonant frequency from the curves is 28,867 rad/s. Obviously, these two frequencies are essentially identical, with the minor deviation attributable to the discrete points connected by the curve in the Bode plot. In conclusion, the correctness of Properties 1 and 2 is confirmed.

(2) In each subplot of Figure 5, when the order $\alpha$ of the fractional-order inductors remains constant and the order $\beta$ of the fractional-order capacitor changes, the five logarithmic magnitude-frequency characteristic curves overlap in the low-frequency band and diverge in the high-frequency band. This suggests that the order $\beta$ of the fractional-order capacitor does not affect the logarithmic magnitude-frequency characteristics of a fractional-order LCL filter in the low-frequency band, but it does have an effect in the high-frequency band. The asymptotic slope of the logarithmic magnitude-frequency characteristic curve of a fractional-order LCL filter in the low-frequency band is $-20\alpha$dB/dec, which is solely related to $\alpha$. For example, when $\alpha$ is 0.8, 1.0, and 1.2, the theoretical low-frequency slopes calculated based on Property 3 are $-16$ dB/dec, $-20$ dB/dec, and $-24$ dB/dec, while the measured values from the curves are $-15.96$ dB/dec, $-19.9$ dB/dec, and $-23.87$ dB/dec, aligning with the theory.

Additionally, the asymptotic slope of the logarithmic magnitude-frequency characteristic curve in the high-frequency band is $-20(2\alpha + \beta)$dB/dec, which is influenced by $\alpha$ and $\beta$. For example, when the combinations of $\alpha$ and $\beta$ are $(0.8, 1.4)$, $(1.0, 1.4)$, and $(1.2, 1.4)$, the theoretical high-frequency slopes are $-60$ dB/dec, $-68$ dB/dec, and $-76$ dB/dec, with the measured curve values being $-60.07$ dB/dec, $-68.01$ dB/dec, and $-76.00$ dB/dec, which demonstrates that the theoretical calculated values are consistent with the actual measured values, too.

In each subplot of Figure 6, when $\beta$ remains constant, and $\alpha$ takes on different values, the slopes of the low-frequency and high-frequency bands of the five logarithmic magnitude-frequency characteristic curves differ, further indicating that the asymptotic slope of the logarithmic magnitude-frequency characteristics of a fractional-order LCL filter in the low-frequency band are solely determined by $\alpha$, while the asymptotic slope in the high-frequency band is influenced by $\alpha$ and $\beta$.

Furthermore, when $\alpha + \beta \in (0, 0.5] \cup [1.5, 2.5] \cup [3.5, 4)$, the corner frequency $\omega_{\mathrm{t}}$ is calculated according to the first formula of (36); for example, when $\alpha = \beta = 1.0$, the theoretically calculated and the measured corner frequencies $\omega_{\mathrm{t}}$ are 28,868 rad/s and 28,867 rad/s, respectively, which are essentially equivalent. When $\alpha + \beta \in (0.5, 1.5) \cup (2.5, 3.5)$, the corner frequency should be calculated according to the second formula of (36); for example, when the combinations of $\alpha$ and $\beta$ are $(0.8, 0.6)$ and $(1.2, 1.4)$, the theoretically calculated corner frequencies $\omega_{\mathrm{t}}$ are 2,024,272 rad/s and 2487 rad/s. The corner frequencies derived from the curve are 2,024,102 rad/s and 2487 rad/s, respectively. This suggests that the theoretically calculated values align with the actual measurements.

To sum up, the correctness of Property 3 has been confirmed.

(3) In each subplot of Figure 5, when $\alpha$ is constant and $\beta$ changes, if $\omega << \omega_{\mathrm{o}}$, the five phase-frequency characteristic curves overlap; if $\omega >> \omega_{\mathrm{o}}$, they diverge. This indicates

that $\beta$ solely influences the phase-frequency characteristics of a fractional-order LCL filter in the high-frequency band but not in the low-frequency band. The low-frequency phase follows the $-\pi\alpha/2$ law. For example, when $\alpha$ is 0.8, 1.0, and 1.2, the theoretically calculated and measured low-frequency phases are $-72°$, $-90°$, and $-108°$, respectively. It can be observed that the low-frequency phase is solely determined by $\alpha$.

Additionally, when $\alpha + \beta \in (0,2]$, the high-frequency phase follows the rule $-(\alpha + \beta/2]\pi$. For example, when the combinations of $\alpha$ and $\beta$ are $(0.8, 0.6)$, $(1.0, 0.6)$, and $(1.2, 0.6)$, both the theoretically calculated and the measured high-frequency phases equal $-198°$, $-234°$, and $-270°$, respectively. When $\alpha + \beta \in (2,4)$, the high-frequency phase follows the rule $-(\alpha + \beta/2)\pi + 2\pi$. For example, when the combinations of $\alpha$ and $\beta$ are $(0.8, 1.4)$, $(1.0, 1.4)$, and $(1.2, 1.4)$, the theoretically calculated and measured high-frequency phases match $90°$, $54°$, and $18°$, respectively. Overall, the correctness of Property 4 has been verified.

(4) The measured value of the phase crossover frequencies $\omega_\mathrm{g}$ shown in Tables 1 and 2 equals the theoretical value calculated by (47). For example, when the combinations of $\alpha$ and $\beta$ are $(0.8, 0.8)$, $(0.8, 1.0)$, $(1.0, 1.0)$, and $(0.6, 1.2)$, the calculated and measured phase-crossing frequencies are equivalent, specifically 508,310 rad/s, 98,864 rad/s, 28,867 rad/s, and 107,916 rad/s, respectively. In conclusion, the correctness of Property 5 is confirmed.

(5) From Table 1 and Figure 5, it is evident that when $\alpha$ remains constant, and $\beta$ varies, the gain crossover frequency $\omega_\mathrm{c}$ remains relatively constant, except when $\alpha + \beta = 2$. For example, when the combinations of $\alpha$ and $\beta$ are $(1.0, 0.8)$ and $(1.0, 1.2)$, the gain crossover frequencies are nearly identical, specifically 1334 rad/s and 1345 rad/s, respectively. Furthermore, when $\alpha + \beta$ approaches 2, there is a resonance in the magnitude-frequency characteristics of a fractional-order LCL filter, resulting in a sudden increase in the gain crossover frequency. Similarly, Table 2 and Figure 6 show that when $\beta$ remains constant and $\alpha$ varies, the gain crossover frequency gradually increases as $\alpha$ decreases, except when $\alpha + \beta = 2$. These simulation results generally align with the observations presented in Figure 5.

Overall, the theoretical derivation and analysis of the frequency characteristics of fractional-order LCL filters have been validated through digital simulations.

## 5. Conclusions

Building on the mathematical model of fractional-order LCL filters, this article deduces five critical properties of these filters, systematically analyzing and summarizing their working principles and frequency characteristics. The primary differences in frequency characteristics between integer-order LCL filters and fractional-order LCL filters are summarized in Table 3.

Compared with an integer-order LCL filter, since the extra parameters of $\alpha$ and $\beta$ are included, although a fractional-order LCL filter is more complicated to analyze, it has many novel behaviors and a wider operating range, which brings great flexibility and advantages to the application. The research in this paper provides a theoretical basis for the design of fractional-order LCL filters. By appropriately selecting the values of $\alpha$ and $\beta$, it is possible to obtain fractional-order LCL filters that can better coordinate filtering performance and system stability. It is expected that fractional-order LCL filters will be applied to replace traditional integer-order LCL filters in power electronic systems, such as inverters and rectifiers, and to construct better-performing fractional-order power electronic systems.

Furthermore, the frequency characteristic analysis method of fractional-order LCL filters proposed in this paper fills the gap in the research on the frequency characteristics of general fractional systems with $2\alpha + \beta$-order (where $\alpha, \beta \in (0, 2)$) and has good theoretical significance.

**Table 3.** The comparison between an integer-order LCL filter and a fractional-order LCL filter.

| Properties | Integer-Order LCL Filters | Fractional-Order LCL Filters | Notes on Fractional-Order LCL Filters |
|---|---|---|---|
| Variables | Three Variables $(L_1, L_2, C)$ | Five Variables $(L_1, L_2, C, \alpha, \beta)$ | $\alpha$ is the order of the fractional-order inductors, and $\beta$ is the order of the fractional-order capacitor. |
| Range of $\alpha$ and $\beta$ | $\alpha = \beta = 1$ | $\alpha, \beta \in (0, 2)$ | An integer-order LCL filter is the special case of a fractional-order LCL filter when $\alpha = \beta = 1$. |
| The transfer function, $G_{gi}(s)$ | $\frac{1}{L_1 L_2 C s}\frac{1}{s^2 + A}$ | $\frac{1}{L_1 L_2 C s^{\alpha}}\frac{1}{s^{\alpha+\beta}+A}$ | |
| Resonance peak | Exists a resonance peak | Exists a resonance peak when $\alpha + \beta = 2$ | The necessary and sufficient condition for the existence of a resonance peak is $\alpha + \beta = 2$. |
| Resonant frequency, $\omega_{rp}$ | $\sqrt{(L_1 + L_2)/(L_1 L_2 C)}$ | $\sqrt{(L_1 + L_2)/(L_1 L_2 C)}$ | $\omega_{rp}$ is determined by the values of $L_1$, $L_2$, and $C$, and is independent of $\alpha$ and $\beta$. |
| Corner frequency, $\omega_t$ | $\sqrt{(L_1 + L_2)/(L_1 L_2 C)}$ | $\begin{cases} \omega_{t1} = \sqrt[\alpha+\beta]{|A\cos[(\alpha+\beta)\pi/2]|} \\ \alpha + \beta \in (0, 0.5] \cup [1.5, 2.5] \cup [3.5, 4) \\ \omega_{t2} = \sqrt[\alpha+\beta]{|A\sin[(\alpha+\beta)\pi/2]|} \\ \alpha + \beta \in (0.5, 1.5) \cup (2.5, 3.5) \end{cases}$ | According to the different value range of $\alpha + \beta$, there are two calculation formulas of $\omega_t$. $\omega_t$ is affected by both $\alpha$ and $\beta$. |
| Slope of the logarithmic magnitude-frequency characteristic, $dL(\omega)/d\lg\omega$ | $\begin{cases} -20\text{ dB/dec}, \omega << \omega_t \\ -60\text{ dB/dec}, \omega >> \omega_t \end{cases}$ | $\begin{cases} -20\alpha\text{dB/dec}, \omega << L\omega_t \\ -20(2\alpha+\beta)\text{dB/dec}, \omega >> \omega_t \end{cases}$ | The slope is only determined by $\alpha$ when $\omega << \omega_t$, while is affected by both $\alpha$ and $\beta$ when $\omega >> \omega_t$. The range of slope is $(0\text{ dB/dec}, -40\text{ dB/dec})$ and $(0\text{ dB/dec}, -120\text{ dB/dec})$ when $\omega << \omega_t$ and $\omega >> \omega_t$, respectively. |
| Center frequency, $\omega_o$ | $\sqrt{(L_1 + L_2)/(L_1 L_2 C)}$ | $\sqrt[\alpha+\beta]{(L_1 + L_2)/(L_1 L_2 C)}$ | $\omega_o$ is affected by both $\alpha$ and $\beta$. |
| Phase-frequency characteristic, $\angle G_{gi}(j\omega)$ | $\begin{cases} -\pi/2, \omega < \omega_o \\ -3\pi/2, \omega > \omega_o \end{cases}$ | $\begin{cases} -\pi\alpha/2, \omega << \omega_o \\ -\pi(\alpha+\beta/2), \alpha+\beta \in (0,2], \omega >> \omega_o \\ -\pi(\alpha+\beta/2)+2\pi, \alpha+\beta \in (2,4), \omega >> \omega_o \end{cases}$ | $\angle G_{gi}(j\omega)$ is only determined by $\alpha$ when $\omega << \omega_o$, while is affected by both $\alpha$ and $\beta$ when $\omega >> \omega_o$. The high-frequency phase curve changes from $-\pi\alpha/2$ to the more lagging direction when $\alpha + \beta \in (0, 2]$, and to the opposite direction when $\alpha + \beta \in (2, 4)$. |
| Phase crossover frequency, $\omega_g$ | $\sqrt{(L_1 + L_2)/(L_1 L_2 C)}$ | $\sqrt[\alpha+\beta]{\dfrac{A\sin[(1-\alpha/2)\pi]}{-\sin[(\alpha+\beta/2)\pi]}}$ | If and only if $\alpha + \beta/2 \in (1, 2)$, the phase-frequency characteristics of a fractional-order LCL filter has a $\omega_g$. |

**Author Contributions:** Conceptualization, J.X.; Methodology, J.X., E.Z. and X.L.; Software, E.Z., G.H. and X.M.; Validation, X.L. and X.M.; Formal analysis, W.L.; Investigation, J.X. and G.H.; Resources, G.H. and W.L.; Data curation, J.X. and E.Z.; Writing—original draft, J.X.; Writing—review & editing, X.L.; Supervision, X.L. All authors have read and agreed to the published version of the manuscript.

**Funding:** This research received no external funding.

**Data Availability Statement:** The original contributions presented in the study are included in the article, further inquiries can be directed to the corresponding authors.

**Conflicts of Interest:** Author Xuanren Meng was employed by the company Electric Power Research Institute of Guangxi Power Grid Corporation. The remaining authors declare that the research was conducted in the absence of any commercial or financial relationships that could be construed as a potential conflict of interest.

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
