# Peer review of "Fractional-Order LCL Filters: Principle, Frequency Characteristics, and Their Analysis"

_fractalfract, doi:10.3390/fractalfract8010038_

Round 1
Reviewer 1 Report
Comments and Suggestions for Authors
Please see the attachment.

There are many grammatical and typo errors in the paper. To make the paper more readable, the author should carefully review it and correct any errors. Also, the equation's alignment must be improved.
Author Response
Dear Reviewer,
We would like to thank you for giving us constructive suggestions which would help us to improve the quality of the paper. The authors have seriously discussed all of these comments. According to the reviewers’ comments, we have tried our best to modify our manuscript to meet the requirements of your journal. In the revised version, changes to our manuscript within the document were all highlighted by using red-colored text.
We are sorry for the inconvenience caused to your work.
We look forward to your positive response.
With kind regards,
Junhua Xu, Ermeng Zeng

Reviewer 2 Report
Comments and Suggestions for Authors
This manuscript aims to investigate the principle, frequency characteristics and analysis of fractional-order LCL (FOLCL) filters. The authors introduce fractional-order inductors and fractional-order capacitors to replace the traditional integer-order components in LCL filters. The authors systematically investigate the principle and frequency characteristics of the FOLCL filter and derive five important properties that provide a theoretical basis for the design and application of FOLCL filters. The study provides a comprehensive analysis of FOLCL filters and offers valuable insights into their design, application and frequency characteristics. However, there are still some issues that should be addressed.
1. The lack of in-depth derivation and analysis. For example, the incorrect expression for the resonant frequency given in Ref. [41] needs further explanation. Does the study of Ref. [41] satisfy property (1)?
2. Equations (24), (25), (27) and (28) and the similar equations in the second case are ambiguous.
The manuscript as a whole needs to be proofread. It contains many linguistic and stylistic errors that need to be corrected. In addition, many sentences are too long, while many paragraphs are too short.
Author Response

(The authors gave the same response as above.)

Round 2
Reviewer 1 Report
Comments and Suggestions for Authors
Accept it present form. Thank You...
Reviewer 2 Report
Comments and Suggestions for Authors
The manuscript is sufficiently improved. I recommend the manuscript for publication in Fractal Fract.